# Causal Intervention for Abstractive Related Work Generation

**Jiachang Liu[1,*], Qi Zhang[2,5,*], Chongyang Shi[1†]**
**Usman Naseem[3], Shoujin Wang[4], Liang Hu[2,5], Ivor W. Tsang[6,7]**
[1]Beijing Institute of Technology, China    [2]Tongji University, China
[3]James Cook University, Australia    [4]University of Technology Sydney, Australia
[5]DeepBlue Academy of Science, China
[6]CFAR, Agency for Science, Technology and Research, Singapore
[7]IHPC, Agency for Science, Technology and Research, Singapore
[1]{jc_liu, cy_shi}@bit.edu.cn    [2]zhangqi_cs@tongji.edu.cn

## Abstract

Abstractive related work generation has attracted increasing attention in generating coherent related work that helps readers grasp the current research. However, most existing models ignore the inherent causality during related work generation, leading to spurious correlations which downgrade the models' generation quality and generalizability. In this study, we argue that causal intervention can address such limitations and improve the quality and coherence of generated related work. To this end, we propose a novel *Causal Intervention Module for Related Work Generation* (CaM) to effectively capture causalities in the generation process. Specifically, we first model the relations among the sentence order, document (reference) correlations, and transitional content in related work generation using a causal graph. Then, to implement causal interventions and mitigate the negative impact of spurious correlations, we use *do*-calculus to derive ordinary conditional probabilities and identify causal effects through CaM. Finally, we subtly fuse CaM with Transformer to obtain an end-to-end related work generation framework. Extensive experiments on two real-world datasets show that CaM can effectively promote the model to learn causal relations and thus produce related work of higher quality and coherence.

## 1 Introduction

A comprehensive related work usually covers abundant reference papers, which costs authors plenty of time in reading and summarization and even forces authors to pursue ever-updating advanced work (Hu and Wan, 2014). Fortunately, the task of related work generation emerged and attracted increasing attention from the community of text summarization and content analysis in recent years (Chen et al., 2021, 2022). Related work generation can be considered as a variant of the multi-document summarization task (Li and Ouyang, 2022). Distinct from multi-document summarization, related work generation entails comparison after the summarization of a set of references and needs to sort out the similarities and differences between these references (Agarwal et al., 2011).

Recently, various abstractive text generation methods have been proposed to generate related work based on the abstracts of references. For example, Xing et al. (2020a) used the context of citation and the abstract of each cited paper as the input to generate related work. Ge et al. (2021) encoded the citation network and used it as external knowledge to generate related work. Chen et al. (2022) proposed a target-aware related work generator that captures the relations between reference papers and the target paper through a target-centered attention mechanism. Equipped with well-designed encoding strategies, external knowledge, or novel training techniques, these studies have made promising progress in generating coherent related work.

However, those models are inclined to explore and exploit spurious correlations, such as high-frequency word/phrase patterns, writing habits, or presentation skills, to build superficial shortcuts between reference papers and the related work of the target paper. Such spurious correlations may harm the quality of the generated related work, especially when distribution shift exists between the testing set and training set. This is because spurious correlations are different from genuine causal relations. They often do not intrinsically contribute to the related work generation and easily cause the robustness problem and impair the models' generalizability (Arjovsky et al., 2019).

Figure 1 illustrates the difference between causality and spurious correlation. The phrases "for example" and "later" are often used to bridge two sentences in related work. Their usage may be attributed to writers' presentation habits about organizing sentence orders or the reference document

---

* The authors contribute equally to this work.
† Corresponding author.

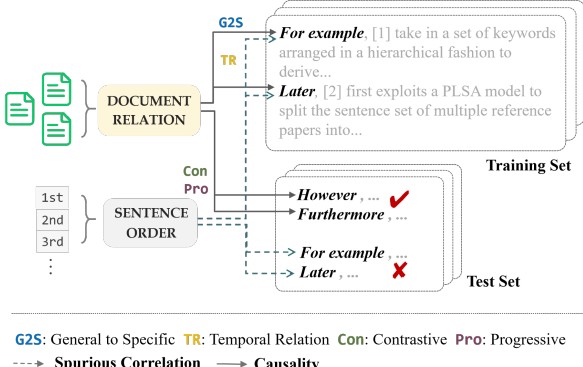

G2S: General to Specific  TR: Temporal Relation  Con: Contrastive  Pro: Progressive
---→ **Spurious Correlation**  ——→ **Causality**

Figure 1: An illustration of the effect difference between causality (solid arrows) and spurious correlations (dashed arrows) in related work generation.

relations corresponding to the sentences. Ideally, a related work generation model is expected to learn the reference relation and distinguish it from the writing habits. However, previous generation models easily capture the superficial habitual sentence organization (a spurious correlation) instead of learning complex causal reference relations, especially when the habitual patterns frequently occur in the training set. In this case, the transitional phrases generated mainly based on writing habits are likely to be unsuitable and subsequently affect the content generation of related work during testing when the training and testing sets are not distributed uniformly.

Fortunately, causal intervention can effectively remove spurious correlations and focus on causal relations by intervening in the learning process. It not only observes the impact of the sentence order and document relation on generating transitional content, but also probes the impact of each possible order on the whole generation of related work, thereby removing the spurious correlations (Pearl, 2009a). Accordingly, causal intervention serving as an effective solution allows causal relations to exert a greater impact and instruct the model to produce the correct content.

Accordingly, to address the aforementioned gaps in existing work for related work generation, we propose a novel **Ca**usal Intervention **M**odule for Related Work Generation (CaM), which effectively removes spurious correlations by performing the causal intervention. Specifically, we first model the relations among sentence order, document relation, and transitional content in related work generation and identify the confounder that raises spurious correlations (see Figure 2). Then, we implement causal intervention that consists of three compo-

nents: 1) *Primitive Intervention* cuts off the connection that induces spurious correlations in the causal graph by leveraging *do*-calculus and *backdoor criterion* (Pearl, 2009a), 2) *Context-aware Remapping* smoothens the distribution of intervened embeddings and injects contextual information, and 3) *Optimal Intensity Learning* learns the best intensity of overall intervention by controlling the output from different parts. Finally, we strategically fuse CaM with Transformer (Vaswani et al., 2017) to deliver an end-to-end causal related work generation model. Our main contributions are as follows:

- To the best of our knowledge, this work is the first attempt to introduce causality theory into the related work generation task.
- We propose a novel **Ca**usal Intervention **M**odule for Related Work Generation (CaM) which utilizes causal intervention to mitigate the impact of spurious correlations. CaM is subtly fused with Transformer to derive an end-to-end causal related work generation model, enabling the propagation of intervened information.
- Extensive experiments on two real-world benchmark datasets demonstrate that our proposed model can generate related works of high quality and verify the effectiveness and rationality of bringing causality theory into the related work generation task.

## 2  Problem Formulation

Given a set of reference papers $D = \{r_1, ..., r_{|D|}\}$, we assume the ground truth related work $Y = (w_1, w_2, ..., w_M)$, where $r_i = (w_1^i, w_2^i, ..., w_{|r_i|}^i)$ denotes a single cited paper, $w_j^i$ is the $j$-th word in $r_i$, and $w_j$ is the $j$-th word in related work $Y$. Generally, the related work generation task can be formulated as generating a related work section $\hat{Y} = (\hat{w}_1, \hat{w}_2, ..., \hat{w}_{\hat{M}})$ based on the reference input $D$ and minimizing the difference between $Y$ and $\hat{Y}$. Considering that the abstract section is usually well-drafted to provide a concise paper summarization (Hu and Wan, 2014), we use the abstract section to represent each reference paper.

## 3  Methodology

We first analyze the causalities in related work generation, identify the confounder that raises spurious correlations, and use a causal graph to model these relations. Then, we implement CaM to enhance the quality of related work through causal intervention. Finally, we describe how CaM, as an intervention

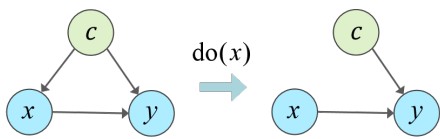

x: document relation   c: sentence order   y: transitional content

Figure 2: Causal graph $G$ for related work generation. By applying $do$-calculus, path $c \rightarrow x$ is cut off and the impact of spurious correlation $c \rightarrow x \rightarrow y$ is mitigated.

module, is integrated with Transformer to intervene in the entire generation process. The overall structure of our model is shown in Figure 3.

### 3.1 Causal Modeling for Related Work Generation

We believe that three aspects play significant roles in related work generation for better depicting the relations between different references, namely, sentence order $c$, document relation $x$, and transitional content $y$ (illustrated in Figure 2). In many cases, sentence order is independent of the specified content and directly establishes relations with transitional content. For example, we tend to use *"firstly"* at the beginning and *"finally"* at the end while composing a paragraph, regardless of what exactly is in between. This relation corresponds to path $c \rightarrow y$, and it should be preserved as a writing experience or habit. Meanwhile, there is a lot of transitional content that portrays the relations between referred papers based on the actual content, at this time, models need to analyze and use these relations. The corresponding path is $x \rightarrow y$.

Though ideally, sentence order and document relation can instruct the generation of transitional content based on practical writing needs, deep learning models are usually unable to trade off the influence of these two aspects correctly but prioritize sentence order. This can be attributed to the fact that sentence order information is easily accessible and learnable. In Figure 2, such relation corresponds to $c \rightarrow x \rightarrow y$. In this case, sentence order $c$ is the confounder that raises a spurious correlation with transitional content $y$. Although performing well on the training set, once a data distribution shift exists between the test set and training set where the test set focuses more on document relations, the transitional content instructed by sentence order can be quite unreliable. To mitigate the impact of the spurious correlation, we need to cut off the path $c \rightarrow x$, enabling the model to generate transitional content based on the correct and reliable causality

of both $c \rightarrow y$ and $x \rightarrow y$.

### 3.2 Causal Intervention Module for Related Work Generation

The proposed CaM contains three parts as shown in Figure 3: **Primitive Intervention** performs causal intervention and preliminarily removes the spurious correlations between sentence order and transitional content.**Context-aware Remapping** captures and fuses contextual information, facilitating the smoothing of the intervened embeddings. **Optimal Intensity Learning** learns the best intensity of holistic causal intervention.

#### 3.2.1 Primitive Intervention

Based on the causal graph G shown in Figure 2, we first perform the following derivation using $do$-calculus and *backdoor criterion*.

$$
\begin{aligned}
p(y|do(x)) &= \textstyle\sum_{\mathbf{c}} p(y|do(x), \mathbf{c})p(\mathbf{c}|do(x)) \\
&= \textstyle\sum_{\mathbf{c}} p(y|x, \mathbf{c})p(\mathbf{c}|do(x)) \qquad (1) \\
&= \textstyle\sum_{\mathbf{c}} p(y|x, \mathbf{c})p(\mathbf{c})
\end{aligned}
$$

In short, the do-calculus is a mathematical representation of an intervention, and the backdoor criterion can help identify the causal effect of $x$ on $y$ (Pearl, 2009b). As a result, by taking into consideration the effect of each possible value of sentence order $c$ on transitional content $y$, $c$ stops affecting document relation $x$ when using $x$ to estimate $y$, which means path $c \rightarrow x$ is cut off (see the arrow-pointed graph in Figure 2). Next, we will explain how to estimate separately $p(y|x, \mathbf{c})$ and $p(\mathbf{c})$ using deep learning models and finally obtain $p(y|do(x))$.

Let $E^{ori} \in \mathbb{R}^{\hat{M} \times d}$ denote the input embeddings corresponding to $\hat{M}$-sized related work and $E^{itv} \in \mathbb{R}^{\hat{M} \times d}$ denote the output embeddings of Primitive Intervention. We first integrate the sentence order information into the input embeddings:

$$
e_i^{odr(j)} = \text{Linear}(e_i^{ori} \oplus o_j), e_i^{ori} \in E^{ori} \qquad (2)
$$

where $j = 1, ..., s$, $s$ is the total number of sentences in the generated related work and $e_i^{odr(j)}$ denotes the order-enhanced embedding for the $i$-th word. We take $o_j = (\lg(j+1), \cdots, \lg(j+1))$ with the same dimension as $e^{ori}$. The linear layer (i.e., $\text{Linear}$) further projects the concatenated embedding to $e^{odr}$ with the same dimension as $e^{ori}$. Accordingly, we have the estimation of $p(y|x, \mathbf{c}) := e^{odr}$. Then, we feed the subsequence

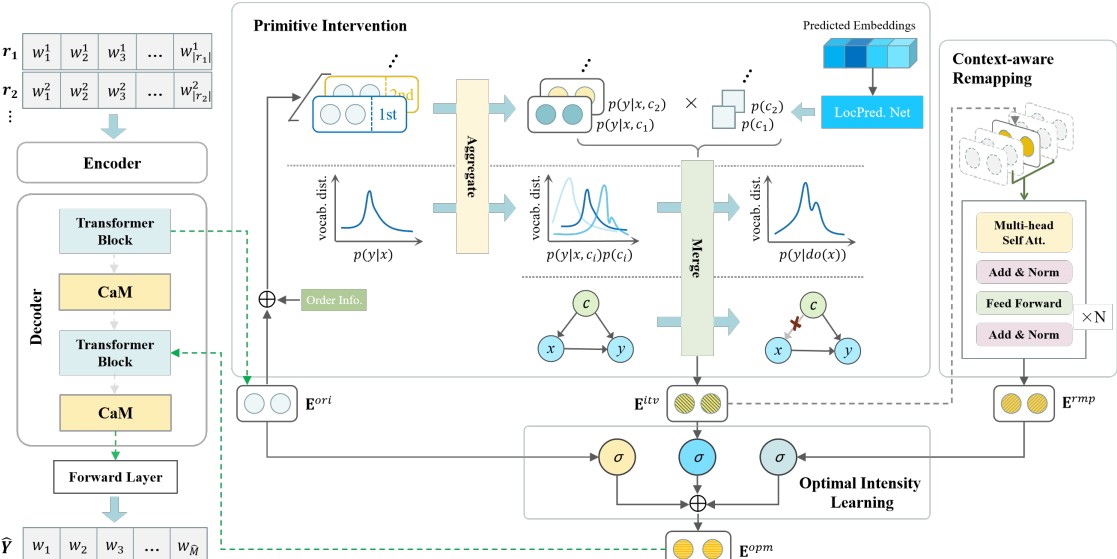

Figure 3: The structure of CaM fused with the Transformer in the decoder. CaM consists of three parts: Primitive Intervention, Context-aware Remapping and Optimal Intensity Learning.

$E_{1:i-1}^{itv}$ to a feed-forward network to predict the sentence position probability of the current decoding word:

$$h_i = \text{Softmax}(\text{FFN}(\text{ReLU}(\sum^{i-1} E_{1:i-1}^{itv})))$$
(3)

where each $h_i^j \in h_i$ denotes the probability. Thus, we estimate the sentence position probability of each decoding word $p(\mathbf{c}) := h$. After obtaining the estimation of $p(y|x, \mathbf{c})$ and $p(\mathbf{c})$, the final embedding with primitive causal intervention is achieved:

$$e_i^{itv} = \sum_{j=1}^{s} e_i^{odr(j)} \times h_i^j, h_i^j \in h_i$$
(4)

where $e_i^{odr(j)} \times h_i^j$ multiplying sentence order probability with order-enhanced embeddings is exactly $p(y|x, \mathbf{c})p(\mathbf{c})$ in Equation 1. Since most transitions are rendered by start words, our CaM intervenes only with these words, namely part of $e^{itv} \in E^{itv}$ being equal to $e^{ori} \in E^{ori}$. For simplicity, we still use $E^{itv}$ in the following.

### 3.2.2 Context-aware Remapping

Two problems may exist in Primitive Intervention: 1) The lack of learnable parameters may lead to the intervened embeddings and the original ones being apart and obstructs the subsequent decoding process. 2) Intervention in individual words may damage the context along with the order-enhanced embedding. To solve the two problems, we propose a Context-aware Remapping mechanism. First, we scan $E^{itv}$ with a context window of fixed size $n_w$:

$$B_i = \text{WIN}(E^{itv})$$
$$= E_{i:i+n_w-1}^{itv}$$
(5)

where $\text{WIN}(\cdot)$ returns a consecutive subsequence of $E^{itv}$ at length $n_w$. Then, we follow the process of Multi-head Attention Mechanism (Vaswani et al., 2017) to update the embeddings in $B_i$:

$$B_i^{rmp} = \text{MultiHead}(B_i, B_i, B_i)$$
$$= (e_i^{rmp}, ..., e_{i+n_w-1}^{rmp})$$
(6)

Even though all embeddings in $B_i$ are updated, we only keep the renewed $e_{i+(n_w/2)}^{rmp} \in B_i^{rmp}$ as the output, and leave the rest unchanged. Since $\text{WIN}(\cdot)$ scans the entire sequence step by step, every embedding will have the chance to update. The output is denoted as $E^{rmp} \in \mathbb{R}^{\hat{M} \times d}$.

### 3.2.3 Optimal Intensity Learning

There is no guarantee that causal intervention with maximum (unaltered) intensity will improve model performance, especially when combined with pre-trained models (Brown et al., 2020; Lewis et al., 2020), as the intervention may conflict with the pre-training strategies. To guarantee performance, we propose the Optimal Intensity Learning.

By applying Primitive Intervention and Context-aware Remapping, we have three types of embeddings, $E^{ori}, E^{itv}$, and $E^{rmp}$. To figure out their respective importance to the final output, we derive the output intensity corresponding to each of them:

$$g^{ori} = \sigma(W^{ori} \cdot e^{ori})$$
(7)
$$g^{itv} = \sigma(W^{itv} \cdot e^{ori})$$
(8)
$$g^{rmp} = \sigma(W^{rmp} \cdot e^{ori})$$
(9)
$$c^{ori}, c^{itv}, c^{rmp} = f_s([g^{ori}, g^{itv}, g^{rmp}])$$
(10)

| Statistic | S2ORC | Delve |
|---|---|---|
| Pairs # | 126k/5k/5k | 72k/3k/3k |
| source # | 5.02 | 3.69 |
| words/sent(doc) # | 1079/45 | 626/26 |
| words/sent(sum) # | 148/6.69 | 181/7.88 |
| vocab size # | 377,431 | 190,381 |

Table 1: Statistics of the datasets

where $\sigma(\cdot)$ is the sigmoid function, $f_s(\cdot)$ is the softmax function. Combining $c^{ori}, c^{itv}, c^{rmp}$, we can obtain the optimal intervention intensity and the final word embedding set $E^{opm} = (e_1^{opm}, ..., e_{\hat{M}}^{opm})$ with causal intervention:

$$e^{opm} = c^{ori}e^{ori} + c^{itv}e^{itv} + c^{rmp}e^{rmp} \quad (11)$$

### 3.3 Fusing CaM with Transformer

To derive an end-to-end causal generation model and ensure that the intervened information can be propagated, we choose to integrate CaM with Transformer (Vaswani et al., 2017). However, unlike the RNN-based models that generate words recurrently (Nallapati et al., 2016), the attention mechanism computes the embeddings of all words in parallel, while the intervention is performed on the sentence start words.

To tackle this challenge, we perform vocabulary mapping on word embeddings before intervention and compare the result with sentence start token [CLS] to obtain $Mask$:

$$I = \text{argmax}[\text{Linear}_{vocab}(E^{ori})] \quad (12)$$

$$Mask = \delta(I, ID_{CLS}) \quad (13)$$

$I$ contains the vocabulary index of each word. $\delta(\cdot)$ compares the values of the two parameters, and returns 1 if the same, 0 otherwise. $Mask$ indicates whether the word is a sentence start word. Therefore, $E^{opm}$ can be calculated as:

$$E^{opm} = E^{opm} \odot Mask + E^{ori} \odot (\sim Mask) \quad (14)$$

The $\odot$ operation multiplies each embedding with the corresponding $\{0, 1\}$ values, and $\sim$ denotes the inverse operation. Note that we omit $Mask$ for conciseness in Section 3.2.3. $Mask$ helps restore the non-sentence-start word embeddings and preserve the intervened sentence-start ones.

As illustrated in Figure 3, we put CaM between the Transformer layers in the decoder. The analysis of the amount and location settings will be discussed in detail in Section 4.6. The model is trained to minimize the cross-entropy loss between the predicted $\hat{Y}$ and the ground-truth $Y$, $v$ is the vocabulary index for $w_i \in Y$:

$$\mathcal{L} = -\sum_i^{\hat{M}} \log p_i^v(\hat{Y}) \quad (15)$$

## 4 Experiments

### 4.1 Datasets

Following the settings in Chen et al. (2021, 2022), we adopt two publicly available datasets derived from the scholar corpora S2ORC (Lo et al., 2020) and Delve (Akujuobi and Zhang, 2017) respectively to evaluate our proposed method in related work generation. S2ORC consists of scientific papers from multiple domains, and Delve focuses on the computer domain. The datasets are summarized in Table 1, where the corresponding ratios of the training/validation/test pairs are detailed [1].

### 4.2 Settings

In our experiments, we incorporate CaM into the Transformer decoder (see Figure 3) and evaluate our model using the resultant encoder-decoder architecture. We utilize pre-trained weights from BERT (Devlin et al., 2019) for both the encoder and decoder of the architecture, as described in Rothe et al. (2020) [2]. Also, when CaM is removed from the decoder in the following experiments, the remaining Transformer model we evaluate still employs pre-trained weights.

In the Transformer architecture we use, the dimension of word embedding is 768, both the number of attention heads and hidden layers in the encoder and decoder are 12, and the intermediate size is 3072. We implement our model with PyTorch on NVIDIA 3080Ti GPU. The maximum reference paper number is set to 5, i.e., $|D| = 5$. We select the first $440/|D|$ words in each reference paper abstract and concatenate them to obtain the model input sequence. The total number of sentences in target related work is set to 6, i.e., $s = 6$. We use beam search for decoding, with a beam size of 4 and a maximum decoding step of 200. We use SGD as the optimizer with a learning rate $1e - 3$. We use ROUGE-1, ROUGE-2, and ROUGE-L on F1 as the metrics (Lin, 2004; Jiang et al., 2022).

---

[1]https://github.com/iriscxy/relatedworkgeneration
[2]https://huggingface.co/docs/transformers/main/en/model_doc/encoder-decoder#transformers.EncoderDecoderModel

| Model | S2ORC | | | Delve | | |
|---|---|---|---|---|---|---|
| | ROUGE-1 | ROUGE-2 | ROUGE-L | ROUGE-1 | ROUGE-2 | ROUGE-L |
| TextRank | 22.36 | 2.65 | 19.73 | 25.25 | 3.04 | 22.14 |
| BertSumEXT | 24.62 | 3.62 | 21.88 | 28.43 | 3.98 | 24.71 |
| MGSum-ext | 24.10 | 3.19 | 20.87 | 27.85 | 3.95 | 24.28 |
| TransformerABS | 21.65 | 3.64 | 20.43 | 26.89 | 3.92 | 23.64 |
| BertSumABS | 23.63 | 4.17 | 21.69 | 28.02 | 3.50 | 24.74 |
| MGSum-abs | 23.94 | 4.58 | 21.57 | 28.13 | 4.12 | 24.95 |
| GS | 23.92 | 4.51 | 22.05 | 28.27 | 4.36 | 25.08 |
| T5-base | 23.20 | 4.01 | 21.41 | 26.38 | 5.69 | 24.35 |
| BART-base | 23.36 | 4.13 | 21.08 | 26.96 | 5.33 | 24.42 |
| longformer | 26.00 | 4.96 | 23.20 | 28.05 | 5.20 | 25.65 |
| RRG | 25.46 | 4.93 | 22.97 | 29.10 | 4.94 | 26.29 |
| NG-Abs | 25.06 | 5.18 | 22.33 | 27.49 | 5.93 | 24.56 |
| TAG | 25.04 | **5.68** | 23.02 | 27.82 | 6.16 | 25.50 |
| **CaM (ours)** | **26.65** | 5.40 | **24.62** | **29.31** | **6.17** | **26.61** |

Table 2: ROUGE scores comparison between our CaM and the baselines.

### 4.2.1 Extractive Methods

(1) **TextRank** (Mihalcea and Tarau, 2004): A graph-based text ranking model that can be used in multi-document sentence extraction. (2) **Bert-SumEXT** (Liu and Lapata, 2019): An extractive summarization model that extends BERT by inserting multiple [CLS] tokens. (3) **MGSum-ext** (Jin et al., 2020): A multi-granularity network that jointly learns different semantic representations.

**Abstractive Methods:** (1) **TransformerABS** (Vaswani et al., 2017): An abstractive summarization model based on Transformer. (2) **Bert-SumABS** (Liu and Lapata, 2019): An abstractive model based on BERT with a designed two-stage fine-tuning approach. (3) **MGSum-abs** (Jin et al., 2020): A multi-granularity interaction network that can be utilized for abstractive document summarization.(4) **GS** (Li et al., 2020): An abstractive summarization model that utilizes special graphs to encode documents to capture cross-document relations. (5) **T5-base** (Raffel et al., 2020): A text-to-text generative language model that leverages transfer learning techniques. (6) **BART-base** (Lewis et al., 2020): A powerful sequence-to-sequence model that combines the benefits of autoregressive and denoising pretraining objectives. (7) **Long-former** (Beltagy et al., 2020): A transformer-based model that can efficiently process long-range dependencies in text. (8) **RGG** (Chen et al., 2021): An encoder-decoder model specifically tailored for related work generation, which constructs and refines the relation graph of reference papers. (9) **NG-Abs** (Zhu et al., 2023): A BART model that optimized jointly with the cross-entropy loss and the proposed differentiable N-gram objectives. (10)

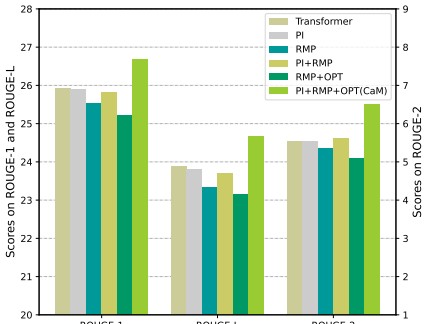

Figure 4: Ablation results on S2ORC.

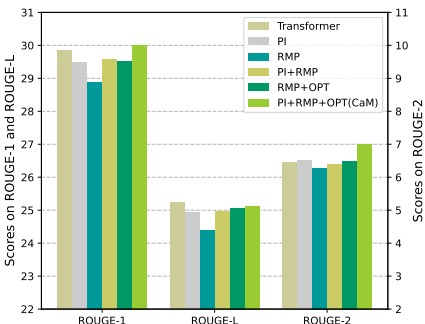

Figure 5: Ablation results on Delve.

**TAG** (Chen et al., 2022): It takes the paper that related work belongs to as the target and employs a target-centered attention mechanism to generate related work.

### 4.3 Overall Performance

It can be found in Table 2 that abstractive models have attracted more attention in recent years and usually outperform extractive ones. Among the generative models, pretrained model T5 and BART achieve promising results in our task without additional design. Meanwhile, Longformer, which is good at handling long text input, also achieves

favorable results. However, the performance of these models is limited by the complexity of the academic content in the dataset.

Our proposed CaM achieves the best performance on both datasets. Due to fusing CaM with Transformer, its large scale ensures that our model can still effectively capture document relations without additional modeling. Accordingly, CaM enables the model to obviate the impact of spurious correlations through causal intervention and promotes the model to learn more robust causalities to achieve the best performance.

## 4.4 Ablation Study

To analyze the contribution of the different components of CaM, we separately control the use of Primitive Intervention (PI), Context-aware Remapping (RMP) and Optimal Intensity Learning (OPT). Figure 4 and Figure 5 show the performance comparison between different variants of CaM.

First, we observe that Transformer already guarantees a desirable base performance. When only PI is used, the model generally shows a slight performance drop. PI+RMP outperforms RMP, showing the necessity of the PI and the effectiveness of RMP. PI+RMP+OPT achieves optimal results, indicating that OPT can effectively exploit the information across different representations.

## 4.5 Human Evaluation

|  | inf | coh | suc | QA |
| --- | --- | --- | --- | --- |
| **CaM** | 2.21 | 2.38 | 2.01 | 41.6 |
| **RRG** | 2.07 | 2.10 | 2.05 | 34.1 |
| **Transformer** | 2.11 | 1.97 | 1.92 | 38.3 |

Table 3: Human evaluation result

We evaluate the quality of related work generated by the CaM, RRG, and Transformer from three perspectives (informativeness, coherence, and succinctness) by randomly selecting forty samples from S2ORC and rating the generated results by 15 PhD students. In the QA task, three PhD students posed three questions for each sample, ensuring that the answers existed in ground truth. Participants need to answer these questions after reading the generated text and we use accuracy as the metric. As table 3 shows, our method achieves the best in informativeness, coherence, and the QA task. However, succinctness is slightly lower than RRG, probably due to the output length limit.

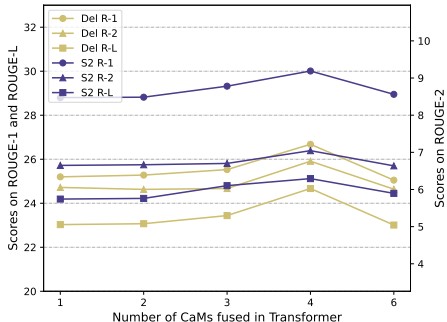

Figure 6: Performance analysis on the number of CaMs fused with Transformer.

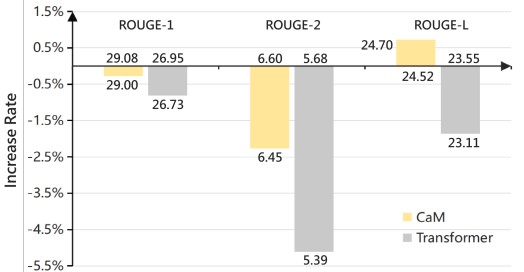

Figure 7: Comparison between Transformer and CaM on original and reordered samples.

## 4.6 Fusing Strategy Comparison

In our setting, the Transformer decoder consists of 12 layers, so there are multiple locations to fuse a different number of CaMs. For each scenario, CaMs are placed evenly among the Transformer decoder layers, and one will always be placed at the end of the entire model. The results of all cases are shown in Figure 6. It can be observed that the model performs best when the number of CaM is 4 both on S2ORC and Delve. With a small number of CaMs, the model may underperform the benchmark model and fail to achieve optimal performance due to the lack of sufficient continuous intervention. If there are too many CaMs, the distance between different CaMs will be too short, leaving an insufficient learning process for the entire fused model, and this might cause the CaMs to bring the noise.

## 4.7 Robustness Analysis

### 4.7.1 Testing with Reordered Samples

We randomly select 50 samples (15 from S2ORC and 35 from Delve) and manually rearrange the order of the cited papers and the order of their corresponding sentences in each sample.Transitional content in related work is also removed since the reordering damages the original logical relations.

Figure 7 shows that CaM has better performance no matter whether the samples have been reordered or not. Regarding the reordered samples, the per-

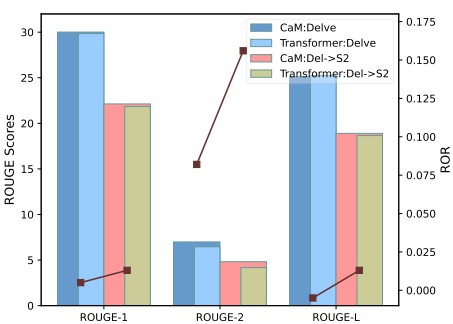

Figure 8: The result of migrating test set from Delve to S2ORC (trained on Delve).

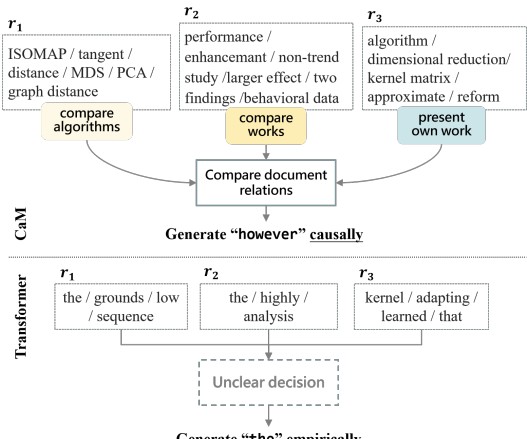

Figure 9: Visualization of the generating process within CaM and Transformer.

formance of Transformer decreases on all three metrics, but CaM only decreases on ROUGE-1 and ROUGE-2 at a much lower rate. Particularly, compared to Transformer, CaM makes improvement on ROUGE-L when tested with reordered samples. The result indicates that CaM is able to tackle the noise disturbance caused by reordering, and the generated content maintains better coherence.

### 4.7.2 Testing with Migrated Test Set

We train the models on Delve and test them on S2ORC, which is a challenging task and significant for robustness analysis. As expected, the performances of all models drop, but we can still obtain credible conclusions. Since CaM outperforms Transformer initially, simply comparing the ROUGE scores after migrating the test set is not informative. To this end, we use *Relative Outperformance Rate* (ROR) for evaluation:

$$\text{ROR} = (\text{S}_{\text{CaM}} - \text{S}_{\text{TF}})/\text{S}_{\text{TF}} \tag{16}$$

$\text{S}_{\text{CaM}}$ and $\text{S}_{\text{TF}}$ are the ROUGE scores of CaM and Transformer, respectively. ROR computes the advantage of CaM over Transformer.

Figure 8 reports that CaM outperforms Transformer regardless of migrating from Delve to S2ORC for testing. In addition, comparing the change of ROR, we observe that although migration brings performance drop, CaM not only maintains its advantage over Transformer but also enlarges it. The above two experiments demonstrate that the CaM effectively learns causalities to improve model robustness.

### 4.8 Causality Visualization

To visualize how causal intervention works in the generation process, we compare the related work generated by Transformer and CaM with a case study. Specifically, we map their cross attention

corresponding to "however" and "the" to the input content using different color shades (Figure 10) to explore what information these two words rely on. More details of the above two experiments can be found in Appendix B.

We picked out the words that "however" and "the" focused on the most and analyzed the implications of these words in the context of the input. The results are shown in Figure 9. It can be found that the words highlighted by CaM have their respective effects in the cited papers. When generating "however", the model aggregates this information, comparing the relations between the documents and producing the correct result. However, there is no obvious connection between the words focused on by Transformer, hence there is no clear decision process after combining the information, and the generated word "the" is simply a result obtained from learned experience and preference. Through causality visualization, it can be observed very concretely how CaM improves model performance by conducting causal intervention.

## 5 Conclusions

In this paper, we propose a Causal Intervention Module for Related Work Generation (CaM) to capture causalities in related work generation. We first model the relations in related work generation using a causal graph. The proposed CaM implements causal intervention and enables the model to capture causality. We subtly fuse CaM with Transformer to obtain an end-to-end model to integrate the intervened information throughout the generation process. Extensive experiments show the superiority of CaM and demonstrate our method's effectiveness.

## Limitations

Although extensive experiments have demonstrated that CaM can effectively improve the performance of the generation model, as mentioned above, since the intervention occurs on the sentence start words, it is inconclusive that CaM can bring improvement if the generation of sentence start words is inaccurate. That is, if it is to be combined with small-scale models without any pre-trained knowledge, then the effectiveness of the model might not be ensured. This will also be a direction of improvement for our future work.

## Acknowledgements

This work is supported by the Fundamental Research Funds for the Central Universities.

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

## A Related Work

### A.1 Related Work Generation

The related work generation task can be viewed as a variant of the multi-document summarization task, and its methods can be categorized as extractive or abstractive. Most of the early studies use extractive methods. The work of Hoang and Kan (2010) is one of the first attempts. They propose a heuristic approach to generate general and specific content separately given a topic tree. Wang et al. (2020) train the model to extract cited text spans through a specific training set and use a greedy algorithm to select the most suitable candidate sentences to compose related work. Most recent studies focus on abstractive approaches. Xing et al. (2020b) use the citation context and the abstract of the cited papers together as inputs to generate citation text. Chen et al. (2021) construct a relation graph of the cited papers during the encoding process and update them iteratively. The relation graph is used as an auxiliary information for decoding. The most recent work is done by Chen et al. (2022), in which they take the paper that related work belongs to as the target and employ a target-centered attention mechanism to generate informative related work.

### A.2 Causal Intervention

In recent years, causality theory has attracted increasing attention in various domains. In the field of recommendation system, Wang et al. (2022a) use the causal graph to model multi-scenario recommendation and solve the problem of existing systems that may introduce unnecessary information from other scenarios.Wang et al. (2022b) propose a framework for sequential recommendation that can perceive data biases by reweighing training data and using inverse propensity scores(Austin, 2011). In the field of natural language processing, Feng et al. (2021) introduce counterfactual reasoning into the sentiment analysis task and leverage the knowledge of both factual and counterfactual samples. Wang and Culotta (2020) propose a method for identifying spurious correlations in the text classification task. The method extracts the words with the highest relevance to the category and uses an estimator to determine whether the correlation is a spurious correlation.

## B Experiment Result for Causal Visualization

In this section, we will give an extra analysis of the experiments introduced in Section 4.8.

### B.1 Generated Related Work Comparison

From Table 4, we can notice that CaM generates enriched content and its meaning is closer to ground truth compared to Transformer. Crucially, when pointing out the problems of previous approaches and presenting the new ones(sentence marked in green), CaM correctly generates "however" at the beginning of the sentence and the entire sentence has a more accurate expression, making the transitions more seamless. But Transformer only generates a very high-frequency word "the" at the same position. It can be perceived that in this process Transformer is not making effective decisions, but simply generating with preference and experience.

### B.2 Visualization Result Analysis on Full Text

Figure 10 visualizes the cross attention of words "however" and "the" in CaM and Transformer. Different cited papers are split with vertical lines. The deeper blue color denotes the higher attention received by the input source word. Judging from the overall coloring situation, we can find that in CaM, there is more deep blue text, as well as more light-colored text. This means the information that "however" focuses on is more targeted and more important, and CaM is capable to produce correct content by accurately capturing document relations and avoid distractions from the confounder. In the result of Transformer, both light and deep blue text become less visible, and the coverage of normal blue increases greatly, indicating that "the" focuses on a wider range of information but lacks emphasis. It indicates that the decision process in Transformer is unclear and ineffective.

Detailed analysis of the exact words they focus on and the decision process of the models is presented in Section 4.8.

| | |
|---|---|
| Ground Truth | Many dimension reduction techniques are proposed based on the vector forms, which are generally divided into two parts, linear and nonlinear. The classical methods of principal component analysis and multi-dimensional scaling are linear, since the outputs returned by these methods are related to the input patterns by a simple linear transformation. **However**, when the input patterns lie on or near a low dimensional sub of the input space, that is the structure of the data set may be highly nonlinear, then linear methods are bound to fail. As the research for manifold learning, several graph-based nonlinear methods have been proposed, such as locally linear em. |
| Transformer | Reduction methods have been proposed on the dimensional space, such are divided into two categories: linear and nonlinear. The first method are the component analysis, the dimensional of the methods are linear to the kernel data. The data of the input dimensional space are not linear to the large dimensional space. **The** data space dimensional of the data be the nonlinear, and are not used. The graph-based nonlinear methods have been proposed. Including as the linear kernel, and entropy. |
| CaM | Reduction methods have been proposed on the kernel space, such as divided into two categories: linear and nonlinear. The first approach component analysis are linear and dimensional analysis are based the kernel of the methods. Data of the input are not represented to the low dimensional space. **However**, the data are not on a low dimensional space. The data space is more nonlinear, and the methods can not be used. The graph-based nonlinear methods have been proposed. Including as the linear entropy. |

Table 4: Related work generated by CaM and Transformer. Analysis of the **bolded** words is in Section 4.8.

**CaM**

**Transformer**

Figure 10: Raw visualization result from CaM on the word "however" and Transformer on the word "the".