# OpenReview forum: "Causal Intervention for Abstractive Related Work Generation"
_EMNLP/2023/Conference — EMNLP 2023 Findings_

### Official Review · Reviewer_WopL · 2023-08-03

**Soundness:** 3

**Excitement:**

2: Mediocre: This paper makes marginal contributions (vs non-contemporaneous work), so I would rather not see it in the conference.

**Paper Topic And Main Contributions:**

The article proposes a novel approach for improving the quality and coherence of abstractive related work generation using causal intervention. The proposed Causal Intervention Module for Related Work Generation (CaM) effectively captures causality in the generation process by mitigating the negative impact of spurious correlations and injecting contextual information. CaM is fused with Transformer to deliver an end-to-end causal related work generation model. Extensive experiments demonstrate that CaM outperforms the state-of-the-art approaches and verifies the effectiveness of bringing causality theory into the related work generation task.

**Questions For The Authors:**

How many different types of causality relations are considered in this paper? Is there any specific design for temporal and logical relations in CAM framework? The dependencies among c, x, and y need to be carefully explained, is there any observations or theoretical evidence to prompt the design of causal graphs?

**Reasons To Accept:**

1. The idea to produce causal intervention for generating related work is very interesting
2. The framework is introduced clearly and sounds reasonable.
3. Proper ablation studies demonstrates the effectiveness.

**Reasons To Reject:**

1. The descriptions of some key concepts need to be clarified, e.g., causality, spurious correlations, and paths.
2. The automatic evaluation of Rouge-based metrics is not convincing enough, more metrics to measure semantic similarity, correctness, distinction, and repetition should be considered.
3. It is a bit overclaimed about the contribution of the causality intervention by cutting off the dependency of spurious correlation, the results show a subtle improvement compared with TAG.

**Reproducibility:**

2: Would be hard pressed to reproduce the results. The contribution depends on data that are simply not available outside the author's institution or consortium; not enough details are provided.

**Reviewer Confidence:**

3: Pretty sure, but there's a chance I missed something. Although I have a good feel for this area in general, I did not carefully check the paper's details, e.g., the math, experimental design, or novelty.

**Typos Grammar Style And Presentation Improvements:**

The writing needs to be improved to provide more details about the motivation and foundation of the proposed approach. Standard formulations and definitions would aid the understanding of concepts.
In Table 2, on S2ORC, the rouge-2 value of CaM is 5.40 which is lower than that of TAG, while CaM is marked as the best model with 5.40 bolded.

---

> ### Author Rebuttal · Authors · 2023-08-29
>
> We much appreciate your time and valuable comments. We provide the following explanations to address your concerns.
>
> **1. The descriptions of some key concepts need to be clarified, e.g., causality, spurious correlations, and paths.**
> > - In our context, causality can be regarded as cause-and-effect relationships between two variables, and there are no confounding variables that could distort the result.
> > - If a correlation between two variables is affected or enhanced by a third variable, then this correlation is a spurious correlation.
> > - A directed path from $a$ to $b$ can be considered as a causal relation without other unobserved variables entangled.
> > - We will provide more explicit definitions of these concepts in the final version.
>
> **2. The automatic evaluation of Rouge-based metrics is not convincing enough, more metrics to measure semantic similarity, correctness, distinction, and repetition should be considered.**
> > - We followed prior related work generation research [1, 2] to choose our automated metrics. Since ROUGE-based metrics mainly measure n-gram content matching, they do lack in the evaluation of aspects such as semantics and repetition.
> > - In addition, we adopt another three metrics, i.e., informativeness, coherence, and succinctness, in human evaluation to alleviate the limitations of the ROUGE metrics to some extent.
> > - To better evaluate the effectiveness of the model, we will add more automated evaluation metrics in the final version.
> > [1] https://aclanthology.org/2021.acl-long.473/
> > [2] https://dl.acm.org/doi/10.1145/3477495.3532065
>
> **3. It is a bit overclaimed about the contribution of the causality intervention by cutting off the dependency of spurious correlation, the results show a subtle improvement compared with TAG.**
> > - Note that in **Table 2**, compared to the TAG, our model achieves significantly higher ROUGE-1 and ROUGE-L scores and gets comparable ROUGE-2 scores on both two datasets. These results indicate the superiority of our proposed CAM and verify the contribution of the causality intervention. In addition, we have provided visualization experiments to interpret the effectiveness of our proposed model, which also verifies the contribution of the causality intervention in improving the interpretability of the related work generation.
>
> **1. How many different types of causality relations are considered in this paper?**
>
> > Two types of causality are considered. They are, respectively, from sentence order to transitional content and from document relations to transitional content.
>
> **2. Is there any specific design for temporal and logical relations in CAM framework?**
>
> > - Among the two causal relations we expect the model to learn, the one from sentence order to transitional content covers a significant amount of temporal relations, since the multi-document summary content organized on the basis of temporal relations tends to be presented according to the order of the current sentence (e.g., using *in early time*, *later*, *recently* for each sentence in order).
> > - The causal relation from document relations to transitional content covers a large number of logical relations since we want the model to decide what transitional content to generate by analyzing the logical relations between documents.
> > - Overall, the design of temporal and logical relations is present in CaM, but not deliberately explicit, and instead causal interventions enable the model to autonomously learn the importance of each of these relations in different scenarios.
>
> **3. The dependencies among c, x, and y need to be carefully explained, is there any observations or theoretical evidence to prompt the design of causal graphs?**
>
> > - As analyzed in [1], if the model's learning process is not interfered with, then the model is prone to learn the correlations that exist in the training data but are unreliable and underperforms on out-of-distribution samples (e.g., since the background of pictures of Huskies is basically snowy, then given a picture of a Chihuahua in the snow, the model will most likely misclassify it as a Husky). Our analysis of the spurious correlations existing in the related work generation process and the modeling of causal graphs are inspired by it to some extent. We hope that causal intervention will guide the model to perform equally well on out-of-distribution samples. Based on this goal and our prior knowledge, we designed the causal graph and derived the causal intervention process in the hope that the model can learn more reliable causal relations motivated by this process.
> > - Although we mentioned that a large number of repeated patterns of transitional words in the training set would cause the model to simply learn a shortcut without analyzing the document relations anymore, we did not have statistics on the real situation in the training set, and we do not think that this is necessary either. On the one hand, when the theoretical modeling is complete, we need the model's generalization ability to learn certain aspects that the theory fails to fully take into account, and on the other hand, detailed ablation experiments can illustrate the effectiveness of our design both theoretically and in terms of implementation details.
> > [1] https://dl.acm.org/doi/10.1145/2939672.2939778
>
> **Typos Grammar Style And Presentation Improvements**
> > - Thanks for your suggestions. We will try to elaborate on our motivation and concepts in our final version.
> > - We are sorry for the second mistake. We will fix it in the final version.
>
> Thank you for your comments and valuable suggestions. We hope our explanations can address all your concerns.

---

### Official Review · Reviewer_DEd8 · 2023-08-03

**Soundness:** 3

**Excitement:**

3: Ambivalent: It has merits (e.g., it reports state-of-the-art results, the idea is nice), but there are key weaknesses (e.g., it describes incremental work), and it can significantly benefit from another round of revision. However, I won't object to accepting it if my co-reviewers champion it.

**Paper Topic And Main Contributions:**

This paper proposed a Causal Intervention Module for Related Work Generation (CaM) to effectively capture causalities in the generation
 process. This model takes the inherent causality during related work generation into consideration, sovling spurious correlations which downgrade the models’ generation quality and generalizability.

**Questions For The Authors:**

1. The authors chose to fuse CaM with Transformer to obtain an end-to-end model, why not other modules. The authors should explain more.
2. As the authors said, CaM is incorprated into the Transformer decoder and utilize pre-trained weights from BERT. As BERT and Transformer has different architectures, how can the pretrained weights of BERT be applied to Transformer?
3. As the authors asid, among the generative models, pretrained model T5 and BART achieve promising results in our task without additional design. why not fuse CAM with T5 or BART?
4. The authors claimed that the proposed CaM achieves the best performance on both datasets. However, as listed in Table 2, ROUGE-2 for TAG on S2ORC is 5.68, higher than 5.40 of CAM.

**Reasons To Accept:**

Extensive experiments on two real-world datasets show that CaM can effectively promote the model to learn causal relations and thus produce related work of higher quality and coherence.

**Reasons To Reject:**

1. The authors chose to fuse CaM with Transformer to obtain an end-to-end model, why not other modules. The authors should explain more.
2. As the authors said, CaM is incorprated into the Transformer decoder and utilize pre-trained weights from BERT. As BERT and Transformer has different architectures, how can the pretrained weights of BERT be applied to Transformer?
3. As the authors said, among the generative models, pretrained model T5 and BART achieve promising results in our task without additional design. Large models may perform better after prompt tuning, why not fuse CAM with T5 or other LLM?
4. The authors claimed that the proposed CaM achieves the best performance on both datasets. However, as listed in Table 2, ROUGE-2 for TAG on S2ORC is 5.68, higher than 5.40 of CAM.

**Reproducibility:**

3: Could reproduce the results with some difficulty. The settings of parameters are underspecified or subjectively determined; the training/evaluation data are not widely available.

**Reviewer Confidence:**

4: Quite sure. I tried to check the important points carefully. It's unlikely, though conceivable, that I missed something that should affect my ratings.

---

> ### Author Rebuttal · Authors · 2023-08-29
>
> Thanks very much for your comments.
>
> **1. The authors chose to fuse CaM with Transformer to obtain an end-to-end model, why not other modules. The authors should explain more.**
>
> > - According to [1], there are over 70 mainstream transformer-based architectures available now. The attention mechanism allows the transformer model to accelerate computation in parallel and far outperforms RNN-based models in terms of both network depth and supported sequence lengths.
> > - As discussed in Limitations, the causal intervention mainly happens on sentence start words. That is, if the sentence start words are not accurately generated in the first place, it is inconclusive that CaM can still bring improvements. Public pre-training research based on transformers is very well developed, and this makes it easier to choose pre-training weights if transformers are used as the backbone model.
> > - The characteristics of transformer layer stacking and the consistency in the dimensionality of input and output information make our packaged CaM easy to integrate with.
>
> > [1] https://arxiv.org/abs/2302.07730
>
> **2. As the authors said, CaM is incorprated into the Transformer decoder and utilize pre-trained weights from BERT. As BERT and Transformer has different architectures, how can the pretrained weights of BERT be applied to Transformer?**
>
> > - As mentioned earlier, we chose the transformer because of its superior performance, more easily integrated structure, and support for pre-trained weights. The pre-trained weights can ensure a relatively favorable generation of initial sentence-initial words, which sets the ground for CaM to exert further effects.
> > - In the experiments, the “Transformer” model in the ablations is a transformer-based encoder-decoder generation model. Specifically, its encoder can be seen as a BERT model without the classification layer, and its decoder can be seen as a BERT model with extra cross-attention layers in each block with the final classification layer being kept. Both the encoder and decoder are loaded with BERT pre-trained weights, and the cross-attention layers are randomly initialized.
> > (ref: https://huggingface.co/docs/transformers/main/en/model_doc/encoder-decoder#transformers.EncoderDecoderModel).
>
> **3. As the authors said, among the generative models, pretrained model T5 and BART achieve promising results in our task without additional design. Large models may perform better after prompt tuning, why not fuse CAM with T5 or other LLM?**
>
> > Since our backbone model can be considered as stacking two BERT-base modules, the total number of parameters is basically the same as the T5-base (220M). We have tried to integrate CaM into T5 or BART decoders, but the improvement is limited. We believe this is because BERT is mainly trained at the token level (masked LM), and our interventions are also performed at the token level, such that the granularity match allows CaM to be more effective. However, the rationale behind the design of T5 or other LLMs is to model multiple types of NLP tasks as sequence-to-sequence tasks, which makes the fine-tuning effect of the model more dependent on the design of the input sequences(including prefix or prompt technique), and the incorporation of certain sub-modules within the model is likely to have a negative effect.
>
> **4. The authors claimed that the proposed CaM achieves the best performance on both datasets. However, as listed in Table 2, ROUGE-2 for TAG on S2ORC is 5.68, higher than 5.40 of CAM.**
>
> > We are sorry for the mistake. We will fix it in the final version.
>
> Thank you for your comments and valuable suggestions. We hope our explanations can address all your concerns.

---

### Official Review · Reviewer_nTMu · 2023-08-12

**Soundness:** 3

**Excitement:**

4: Strong: This paper deepens the understanding of some phenomenon or lowers the barriers to an existing research direction.

**Paper Topic And Main Contributions:**

This paper introduces a new approach to related work generation that separates the spurious correlations associated with sentence order that may confound transitional language (as opposed to understanding nuanced differences in how the works compare to each other).  They introduce the CaM module that does this using primitive intervention and a few other context-related modeling techniques and integrate this within a transformer architecture by alternating with transformer blocks. They achieve improvements in human and automatic metrics on two datasets and perform ablations.  They also demonstrate more robustness to adversarial augmentation (reordering sentences).

**Questions For The Authors:**

The transformer model used in the ablations – I assume this is using the BERT architecture just without the CaM add-ons, and that the “transformer” model was still trained on the same data as the full CaM model with the same settings.  Is that the case?

**Reasons To Accept:**

- Interesting and very novel approach to this problem
- New causal Intervention Module which could potentially have applications in other tasks with similar confounding factors
- Promising quantitative improvements with multiple metrics with ablations and analysis of the results to show the utility of each component and the robustness of the technique.

**Reasons To Reject:**

- Section 3.2.1 and 3.2.2 can be a bit confusing and need to be expanded more (having a running example might help with this).  Some of the variable naming conventions were difficult to keep track of (especially when there are multiple different subscripts and superscripts) .
- Only one experiment is done with human evaluations and the performance differences are much smaller in the human evals.  Expanding the human evaluation would strengthen the results
- Discussion of results felt a bit condensed.  I would love to see this expanded on.

**Reproducibility:**

3: Could reproduce the results with some difficulty. The settings of parameters are underspecified or subjectively determined; the training/evaluation data are not widely available.

**Reviewer Confidence:**

3: Pretty sure, but there's a chance I missed something. Although I have a good feel for this area in general, I did not carefully check the paper's details, e.g., the math, experimental design, or novelty.

**Typos Grammar Style And Presentation Improvements:**

Figure 8, the rouge&ROR on the same axis was a bit hard to read.  I might suggest just an ror graph would be better with the rouge numbers pushed to the appendices.

---

> ### Author Rebuttal · Authors · 2023-08-29
>
> We sincerely appreciate you taking the time and providing valuable comments. In the following, we provide a detailed response to address all of your concerns.
>
> **1. Sections 3.2.1 and 3.2.2 can be a bit confusing and need to be expanded more (having a running example might help with this). Some of the variable naming conventions were difficult to keep track of (especially when there are multiple different subscripts and superscripts).**
>
> > + Thanks very much for your suggestion. Section 3.2.1 and Section 3.2.2 introduce the key parts of the causal intervention module. To provide a better understanding of the two sections, we will provide examples based on Figure 1 or Figure 2 in Sections 3.2.1 and 3.2.2.
> > + Regarding the variable naming conventions, we will carefully check and refine the notations to guarantee consistency in the paper. In addition, we have substituted the complex superscripts and subscripts with single characters, for example changing **ori, itv, rmp, and opm** with **b, a, c, o**, to provide clear/simplified formations of our proposed model.
>
> **2. Only one experiment is done with human evaluations and the performance differences are much smaller in the human evals. Expanding the human evaluation would strengthen the results. Discussion of results felt a bit condensed. I would love to see this expanded on.**
>
> > + Note that in this paper, we conducted human evaluation from three aspects, informativeness, coherence, and succinctness [1,2]. Our model scores the highest point on informativeness and coherence, and just a little bit lower on succinctness compared to RRG (probably due to the max output length).
> > + To address your concern, we expanded the human evaluation and designed an additional QA task. Specifically, three Ph.D. students posed three questions for each of the sampled 40 instances, ensuring that the answers existed in the ground truth. We invited participants to answer these questions after reading the text generated by CaM, RRG, and Transformer, and the PhD students would judge the correctness. The result precision for each of the three models are as follows:**CaM-41.6%, Transformer-38.3%, RRG-34.1%**
> > We will update the human evaluation in the final version.
>
> [1] https://aclanthology.org/2021.acl-long.473/
>
> [2] https://dl.acm.org/doi/10.1145/3477495.3532065
>
> **3. Discussion of results felt a bit condensed. I would love to see this expanded on.**
>
> > Thanks for your suggestion. We will polish the discussion of our experimental results in the final version to deliver a high-quality evaluation analysis.
>
> **Q1. The transformer model used in the ablations – I assume this is using the BERT architecture just without the CaM add-ons,  and that the “transformer” model was still trained on the same data as the full CaM model with the same settings. Is that the case?**
>
> > - Yes. The “Transformer” model in the ablations is the one without CaM add-ons, but it is a transformer-based encoder-decoder generation model and is not exactly the same as BERT. Specifically, the encoder can be seen as a BERT model without the classification layer, and its output is the input of the decoder. The decoder can be seen as a BERT model with extra cross-attention layers in each block, and the final classification layer is kept. Both the encoder and decoder are loaded with BERT pre-trained weights, and the cross-attention layers are randomly initialized
> (ref: https://huggingface.co/docs/transformers/main/en/model_doc/encoder-decoder#transformers.EncoderDecoderModel).
> > - We have tried to integrate CaM into T5 or BART decoders, but the improvement is limited. We believe this is because BERT is mainly trained at the token level (masked LM), and our interventions are also performed at the token level, such that the granularity match allows CaM to be more effective.
> > - The rationale behind the design of T5 or other LLMs is to model multiple types of NLP tasks as sequence-to-sequence tasks, which makes the fine-tuning effect of the model more dependent on the design of the input sequences (including prefix or prompt technique), and the incorporation of certain sub-modules within the model is likely to have a negative effect.
>
> **Typos Grammar Style And Presentation Improvements**
>
> > Thanks for your valuable suggestions and comments. We will redraw Figure 8 to report the results of ROR and draw another graph to report the results of ROUGE in the appendices.

---

### Meta-Review · Area_Chair_V6AU · 2023-09-11

**Recommendation:** 3

**Metareview:**

The paper improves the performance of related work generation by introducing casual intervention that addresses specific confounds (e.g. sentence order and transitional languages) to transformer-based pretrained models. Experiments demonstrate that it works well compared to state-of-the-art models. Most reviewers thought the proposed approach is novel and interesting, and that the experiments are extensive in that it covers two datasets and there are various tests and analyses to measure the utility of each component. That said, there are some concerns that reviewers have identified:  (1) from the discussions it appears that the proposed casual intervention technique mostly benefits BERT-based models and not other pretrained models - while the authors had a reasonable speculation (owing to the differences in their pretraining objectives), this is a significant limitation that needs to be properly discussed in the paper; (2) reviewers would appreciate if the human evaluation is done more thoroughly; (3) the performance gain appears to be less than what it claims - authors should probably tone this down to avoid misleading the readers; and (4) reviewers thought certain parts of the paper are unclear or difficult to follow (e.g. key concepts and the baseline transformer encoder-decoder model needs more clarification).

---

### Decision · Program_Chairs · 2023-10-07

**Decision:**

Accept-Findings

**Comment:**

The paper improves the performance of related work generation by introducing casual intervention that addresses specific confounds (e.g. sentence order and transitional languages) to transformer-based pretrained models. Experiments demonstrate that it works well compared to state-of-the-art models. Most reviewers thought the proposed approach is novel and interesting, and that the experiments are extensive in that it covers two datasets and there are various tests and analyses to measure the utility of each component. That said, there are some concerns that reviewers have identified:  (1) from the discussions it appears that the proposed casual intervention technique mostly benefits BERT-based models and not other pretrained models - while the authors had a reasonable speculation (owing to the differences in their pretraining objectives), this is a significant limitation that needs to be properly discussed in the paper; (2) reviewers would appreciate if the human evaluation is done more thoroughly; (3) the performance gain appears to be less than what it claims - authors should probably tone this down to avoid misleading the readers; and (4) reviewers thought certain parts of the paper are unclear or difficult to follow (e.g. key concepts and the baseline transformer encoder-decoder model needs more clarification).